# The Contribution of Psychological Factors to Inter-Individual Variability in Conditioned Pain Modulation Is Limited in Young Healthy Subjects

**DOI:** 10.3390/brainsci12050623

**Published:** 2022-05-10

**Authors:** Philipp Graeff, Regina Stacheneder, Laura Alt, Ruth Ruscheweyh

**Affiliations:** 1Graduate School of Systemic Neuroscience, Ludwig-Maximilians-University Munich, 82152 Planegg, Germany; ruth.ruscheweyh@med.uni-muenchen.de; 2RTG 2175 “Perception in Context and Its Neural Basis”, Ludwig-Maximilians-University Munich, 82152 Planegg, Germany; 3Department of Neurology, University Hospital Großhadern, Ludwig-Maximilians-University Munich, 81377 Munich, Germany; regina.stacheneder@gmail.com (R.S.); laura.alt@gmx.net (L.A.); 4Department of Neurology, Ulm University Hospital, 89081 Ulm, Germany

**Keywords:** conditioned pain modulation, endogenous analgesia, inter-individual differences, psychological factors, CPM variability

## Abstract

Conditioned pain modulation (CPM) describes the decrease in pain perception of a test stimulus (TS) when presented together with a heterotopic painful conditioning stimulus (CS). Inter-individual differences in CPM are large and have been suggested to reflect differences in endogenous pain modulation. In a previous analysis, we demonstrated that in young, healthy participants, inter-individual differences account for about one-third of CPM variance, with age and sex together explaining only 1%. Here, we investigated if psychological factors explain significant amounts of inter-individual variance in CPM. Using the same dataset as before, we performed both cross-sectional (*n* = 126) and repeated measures (*n* = 52, 118 observations) analysis and the corresponding variance decompositions, using results of psychological questionnaires assessing depression, trait anxiety and pain catastrophizing. Psychological factors did not significantly predict CPM magnitude, neither directly nor when interactions with the CPM paradigm were assessed; however, the interaction between depression and the paradigm approached significance. Variance decomposition showed that the interaction between depression and the CPM paradigm explained an appreciable amount of variance (3.0%), but this proportion seems small when compared to the residual inter-individual differences (35.4%). The main effects of the psychological factors and the interactions of anxiety or catastrophizing with the CPM paradigm are explained at <0.1% each. These results show that the contribution of psychological factors to inter-individual CPM differences in healthy participants is limited and that the large inter-individual variability in the CPM effect remains largely unexplained.

## 1. Introduction

Conditioned pain modulation (CPM) describes a phenomenon of human endogenous pain inhibition thought to be the psychophysical equivalent to the “diffuse noxious inhibitory controls” (DNIC) described in animal experiments [1]. During CPM testing, a noxious test stimulus (TS) is presented in parallel with, or directly after, a heterotopic noxious conditioning stimulus (CS), with the underlying principle being summarized as “pain inhibits pain” [2]. When presented with the CS, the TS is perceived as less painful compared to the presentation without CS [1]. Inter-individual differences in CPM magnitude are substantial and can predict an individual’s susceptibility to acute or chronic pain [3,4]. The basis for these inter-individual differences has not yet been understood, although age and sex may make a contribution [5,6]. In addition to inter-individual differences, experimental factors such as CS intensity and the CPM paradigm may also influence the CPM effect [7,8].

In a previous study on healthy young individuals, we used repeated measures analysis to identify the amount of variance in the CPM effect explained by the inter-individual differences (above age and sex), the experimental factors CPM paradigm, CS intensity and measurement repeat. It resulted that residual inter-individual differences accounted for 34.2% while age and sex accounted for only 1.0% and the other experimental factors together explained 10.5% of the variance [9]. This shows that inter-individual differences in the CPM effect are large and largely unexplained.

Psychological factors might explain a part of these residual inter-individual differences. Pain perception has been shown to increase with higher scores of anxiety, depression and pain catastrophizing [10,11,12]. However, discrepancies in the literature exist, with other studies finding no such association [13,14,15]. CPM is reduced in a variety of chronic pain conditions [16], which often shows increased scores for depression, anxiety and catastrophizing [3,4,5]. Therefore, one could hypothesize that increased scores for these psychological factors could be associated with a decreased CPM effect. A previous meta-analysis of cross-sectional data did not find an overall relation between CPM magnitude and psychological factors but found paradigm-specific relations, i.e., of depression with heat-based CPM [17]. Repeated measures investigations may increase sensitivity by reducing the influence of session-specific factors and can provide direct information on how much of the inter-individual variance in CPM is explained by psychological factors. In addition, discerning the contribution of psychological factors to inter-individual CPM variance in a healthy population may establish a normative baseline to which their effect in chronic pain populations can be compared.

We, therefore, followed up on our previous analysis [9] and used repeated measures analysis to investigate if depression, anxiety and pain catastrophizing scores explained a significant amount of the inter-individual differences in the CPM effect. For this, we used a subsample of our previous cohort, for which all three scores were present.

## 2. Materials and Methods

### 2.1. Pooled Data

Data were pooled from the same seven studies used previously [9], resulting in 126 participants for cross-sectional analysis and 52 participants (with 118 observations) for repeated measures analysis. Only participants with Beck’s Depression Inventory (BDI [18]), State-Trait Anxiety Inventory (trait subscale, STAI-T [19]) and Pain Catastrophizing Scale (PCS [20]) scores available (collected during the first session of the respective study) were included. These are three self-rating questionnaires that are reliable and valid for the assessment of depression, anxiety and pain catastrophizing and are widely used in pain research.

Details of CPM measurement are reported in [9]. Briefly, the conditioning stimulus (CS) was hand immersion into cold water for 60–120 s, targeting a pain intensity ≥3 on a 10-point NRS (0 = no pain, 10 = strongest pain imaginable). The test stimulus (TS) was either contact heat for 60–90 s or electrical stimulation of the sural nerve. TS was applied once alone and once during the CS. The three paradigms used were (TS/CS): (1) electrical/120 s cold, (2) 60 s heat/90 s cold, and (3) 30 s heat/60 s cold. The repeated measures sample included only paradigms 1 and 3.

The CPM effect was calculated as the percentage difference between the test stimulus rating at baseline (**NRS**_TS(baseline)_) and during conditioning stimulation (**NRS**_TS(cond)_), where a more negative result denotes a stronger CPM effect:$$\mathrm{CPM}\;\mathrm{effect}=\frac{{NRS}_{\mathrm{TS}(\mathrm{cond})}-{NRS}_{\mathrm{TS}\left(\mathrm{baseline}\right)}}{{NRS}_{\mathrm{TS}\left(\mathrm{baseline}\right)}}\times 100$$

### 2.2. Statistical Models

For cross-sectional analysis, linear regression models were constructed using the *lm()* function of the *stats* package in R [21]. For repeated measures analysis, mixed models were constructed using the *lmer()* function of the *lme4* package [22]. Based on our previous research, we included age, sex, CS temperature, measurement repeat and the CPM paradigm in the models [9].

We first investigated the main effect of the psychological factors in a linear model (Model 1). For repeated measures analysis, we constructed mixed models with and without the inclusion of psychological factors as the main effects, to allow for a comparison of explained variance (Models 2 and 3). (1|participant) denotes the participant as a random effect.
CPM effect ~ CS_temp_ + age + sex + BDI + STAI Trait + PCS + paradigm (1)
CPM effect ~ CS_temp_ + age + sex + paradigm + repeat + (1|participant) (2)
CPM effect ~ CS_temp_ + age + sex + BDI + STAI Trait + PCS + paradigm + repeat + (1|participant) (3)

In the next step, we included interaction terms between the CPM paradigm and the psychological factors in both the cross-sectional (linear) and repeated measures (mixed model) analyses. Note, however, that in R notation, ‘*’ indicates that both the main effects and the respective interactions are included in the model.
CPM effect ~ CS_temp_ + age + sex + BDI*paradigm + STAI Trait*paradigm + PCS*paradigm (4)
CPM effect ~ CS_temp_ + age + sex + BDI*paradigm + STAI Trait*paradigm + PCS*paradigm + repeat + (1|participant)(5)

Significance was tested by the *lm()* function (linear models), and by the *Anova()* function (Wald’s chi-square test) (*car* package [23]) for mixed models. A *p* < 0.05 was considered significant.

Variance decomposition of the repeated measures analysis was performed as described in detail in our previous article [9]. Briefly, we used the *r.squaredGLMM()* function of the *MuMIn* package [24] on the mixed Models 2, 3 and 5 to determine the variance explained by the fixed effects and residual inter-individual variance, and the *calc.relimp()* function of *relaimpo* package [25] on the following linear models to further decompose the fixed effects variance of Models 3 and 5:CPM effect ~ CS_temp_ + age + sex + BDI + STAI Trait + PCS + paradigm + repeat(6)
CPM effect ~ CS_temp_ + age + sex + BDI*paradigm + STAI Trait*paradigm + PCS*paradigm + repeat(7)

## 3. Results

In the cross-sectional sample (*n* = 126, 91 females), the mean age was 29 ± 12 years and the mean CPM effect was significant at −19.5 ± 25.9% (*p* < 0.001). Mean BDI, STAI Trait and PCS scores were 3 ± 4 (range: 0–17), 37 ± 9 (21–56) and 14 ± 9 (0–33), respectively. In the repeated measures sample (52 subjects/28 females, 118 experiments), the mean age was 24 ± 6 years and the mean CPM effect was −16.9 ± 21.2% (*p* < 0.001). Mean BDI, STAI Trait and PCS scores were 4 ± 4 (range: 0–17), 38 ± 9 (21–56) and 14 ± 8 (0–33), respectively.

### 3.1. Main Effects and Interactions

In the cross-sectional sample, none of the psychological factors was a significant predictor of CPM (Model 1, Table 1). CS physical intensity, age, CPM paradigm or sex were also non-significant.

The repeated measures analysis also revealed no significant main effect of any psychological factor on the CPM effect (Model 3, Table 2). As in our previous analysis [9], CS physical intensity (i.e., cold water bath temperature) was a significant predictor of CPM size in both Models 2 and 3 (both *p* < 0.01), but age, sex, paradigm or measurement repeat were not significant (Table 2).

The results of Nahman-Averbuch et al. [17] prompted us to look for paradigm-specific relations between the CPM effect and psychological factors. However, in our cross-sectional analysis (Model 4) none of the psychological factors exhibited a significant interaction with the paradigm (Appendix A). Similarly, repeated measures analysis (Model 5) showed no significant interactions. Of the three interactions tested, paradigm*BDI was the largest, although non-significant at *p* = 0.130 (Appendix A).

### 3.2. Analysis of Explained Variance

In a complementary approach, we investigated the possible contribution of psychological factors to inter-individual CPM differences through analysing the explained variance by adding psychological factors or their interactions to the models. A meaningful effect of psychological factors on CPM magnitude would be expected to show as an increase in the fixed effects variance with a parallel decrease in the residual inter-individual variance. Including psychological factors as main effects (Model 2 vs. Model 3) did not increase the variance explained by the fixed effects (11.5% vs. 11.6%) and increased (rather than decreased) the variance explained by residual inter-individual effects (34.2% vs. 36.2%).

However, including the interactions between psychological factors and the CPM paradigm (Model 5 vs. Model 3) resulted in an appreciable increase in fixed effects variance (14.3% vs. 11.6%), combined with a small decrease in residual inter-individual variance (35.4% vs. 36.2%, Figure 1). This suggests that paradigm-specific interactions can explain some of the inter-individual variances of the CPM effect. To determine which interaction(s) were responsible for this change, we decomposed fixed effect variance contributions (Model 7, Appendix A). This revealed that only the BDI*paradigm interaction closely approached significance (*p* = 0.053), while the interactions with STAI Trait or PCS were non-significant at *p* = 0.708 and *p* = 0.561, respectively. Indeed, BDI*paradigm explained 3.0% of the fixed effects variance, while CS physical intensity explained 4.5%, followed by the CPM paradigm (2.1%) and measurement repeat (1.0%) (Figure 1 and Appendix A).

When investigating the direction of the interaction, we found a decrease in the CPM effect with increased BDI values in the electrical/120 s cold paradigm (coefficient = 0.16) and the opposite in the 30 s heat/60 s cold paradigm (coefficient = −0.18, Appendix A).

## 4. Discussion

This follow-up investigation shows that in young healthy subjects:(i)Psychological factors, such as depression, anxiety or pain catastrophizing, do not significantly predict the CPM effect when different CPM paradigms are pooled.(ii)Depression can explain some amount of inter-individual CPM variance dependent on the CPM paradigm. However, this contribution remains small (3.0%) when compared to the residual inter-individual variance (35.4%).

Our previous investigation [9] showed that inter-individual differences account for approximately one-third of the variance in CPM magnitude, with age and sex contributing only ~1% combined. In the present analysis, we set out to determine if part of the residual, unexplained inter-individual differences can be explained by psychological factors. Indeed, it has been shown before that pain perception and some measures of endogenous pain modulation may be dysregulated (i.e., increased pain perception and/or reduced endogenous pain inhibition) in populations suffering from depressive [11,26] or anxiety disorders [10], or when healthy subjects engage in acute catastrophizing thoughts [27] or experience unpleasant emotions or fear [28,29]. However, it must be mentioned that other studies find no such relationship between pain perception and anxiety [14], catastrophizing [15] and depression [13] in healthy individuals.

In the present analysis, depression, trait anxiety and pain catastrophizing scores did not significantly predict the CPM effect, neither in the cross-sectional nor in the more sensitive repeated measures analysis. A previous meta-analysis of cross-sectional studies [17], investigating the association of various psychological scores with CPM magnitude, also found no overall effect. As our analysis contained different CPM paradigms (involving different modalities as test stimuli) and the previous meta-analysis found paradigm/modality-specific relations between psychological factors and CPM magnitude [17], we investigated if including the interaction between psychological factors and the CPM paradigm significantly improved our models. This was not the case, although, in the mixed model analysis, the interaction between depression scores and the CPM paradigm approached significance.

One advantage of repeated measures analysis is that novel techniques [30,31] allow to estimate the contribution of (residual) inter-individual differences to a variable—in this case, CPM magnitude—and compare it to the variance explained by known (fixed) effects. It turned out that the interaction between depression scores and the CPM paradigm increased the variance explained by the fixed effects while decreasing the residual inter-individual variance. The interaction term again closely approached significance. When analysing the direction of interaction, the CPM effect decreased with increased depression scores in the electrical/cold paradigm, while the effect was opposite in the heat/cold paradigm.

These results support the previous findings [17] that the relation between CPM magnitude and psychological factors can be dependent on the CPM paradigm (especially on test stimulus modality). However, the specifics of the single interactions were different, as the previous study found a significant positive relation between depression and the CPM effect (i.e., more depression, less effective CPM) when heat was used as the TS. We found a positive relationship between depression and electrical CPM (i.e., more depression, less effective CPM), while the relation was negative between depression and heat CPM (i.e., more depression, more effective CPM). In addition, we did not find an interaction between pain catastrophizing and the CPM paradigm, while the previous study found such a relation when electrical pain was used as TS. The specific methodological differences of the CPM paradigms used may contribute to these differences.

Together, the present and previous [9,17] results emphasize that the CPM paradigm can make an important contribution to CPM magnitude, not only directly, but possibly also by its interaction with psychological factors. However, it must be recognized that the additional contribution of the psychological factors investigated here and their interactions with the CPM paradigm to the variance explained was small (3.0%) when compared to the residual inter-individual variance (35.4%). Therefore, further investigations will be necessary to address the basis of the large inter-individual differences in CPM magnitude.

### 4.1. Future Directions

Several additional factors could contribute to inter-individual differences in CPM. First, there may be other psychological factors, transient or not, that could influence the CPM effect. For example, active cognitive strategies have been shown to influence descending pain inhibition [32,33], and intrinsic attention to pain, i.e., an individual’s tendency to attend to painful stimuli is related to CPM [34]. Stressful tasks have also been shown to inhibit the CPM effect [35]. Expectations towards the direction and magnitude of CPM may be individually different and can affect CPM magnitude [36]. Second, allelic differences in certain genes have an effect on both pain perception and CPM [37,38]. Third, Ibancos-Losada et al. [39] suggested that individual differences in perceived unpleasantness of certain pain modalities over others may influence their CPM effect. Fourth, cardiovascular reactivity to pain may also affect CPM [40]. There may be many more factors not mentioned here. It remains to be determined how much of the inter-individual differences in the CPM effect are explained by these factors, both alone and in combination. Repeated measures analysis with a determination of explained variance, as performed here, may aid to perform these investigations.

In addition, it is possible that we found little relation between psychological factors and the CPM effect because our study population was healthy. CPM may not be affected by psychological factors if they are within a fairly narrow and low range, as they were within our study. However, this is a necessary first step in investigating the relation between psychological factors and CPM and also to establish a normative baseline for a healthy population. The next steps will include the investigation of clinical populations. A previous meta-analysis also suggested no association between the CPM effect and psychological factors in chronic pain patients [17]. However, the situation is complex. Chronic pain patients often have psychological comorbidities, resulting in elevated scores for depression, anxiety and/or catastrophizing [3,4,5], and they also exhibit a relationship between these scores and increased pain perception and/or a reduced CPM effect [41,42,43]. Therefore, to dissect the relation between CPM, psychological factors and chronic pain, at least three different groups of patients will have to be compared: patients with chronic pain but without psychological comorbidity, patients with increased depression, anxiety and/or catastrophizing scores but without chronic pain, and patients with chronic pain and psychological comorbidities. Moreover, chronic pain populations can be very heterogeneous regarding the type and cause of chronic pain. Limiting investigation to one of the major types of chronic pain known to be associated with reduced CPM effect might be a good starting point.

Moreover, our present and previous [9,17] data show that the CPM paradigm has an effect on CPM magnitude and also on the relation between CPM and psychological factors. Future studies should take this into account and either use a single paradigm or ideally compare multiple paradigms, including paradigms not examined in this study, e.g., pressure pain.

### 4.2. Strengths and Limitations

The most important strength of our study is that it used repeated measures analysis to directly assess the CPM variance accounted for by inter-individual differences. An important limitation of our study is that the number of included subjects and experiments was limited because not all subjects included in our previous investigation had psychological scores available. This might have been the reason that only a trend but no significance was found for the interaction between depression and the CPM paradigm. Moreover, results cannot be generalized to chronic pain patients, who exhibit a broader range of psychological scores and will need to be studied separately (see Section 4.1). In addition, our study only examined two CPM paradigms. The addition of a pressure pain paradigm could help further investigate the paradigm-specific interactions with psychological factors.

## 5. Conclusions

The psychological factors depression, anxiety and pain catastrophizing did not make a significant contribution to explaining inter-individual variance in the CPM effect of healthy young subjects. Interaction analysis suggested that depression scores may have a modality-specific effect on CPM (*p* = 0.053). However, compared to the residual inter-individual variance (35.4%), the variance explained by the interaction between depression and the CPM paradigm was small (3.0%), as was the variance explained by age and sex (<1%). In conclusion, up to now, most of the inter-individual variance in the CPM effect remains unexplained.

## Figures and Tables

**Figure 1 brainsci-12-00623-f001:**
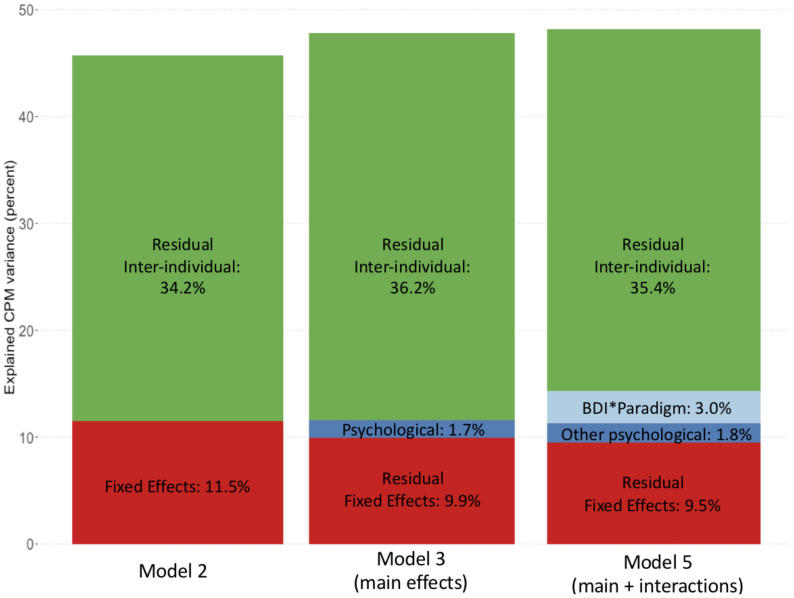
Variance decomposition while adding psychological factors to the model. Variance in CPM magnitude explained by Model 2 (including age, sex, repeat, CPM paradigm and CS intensity, but no psychological factors), Model 3 (additionally including psychological factors as main effects), and Model 5 (additionally including interactions of psychological factors with CPM paradigm). “*” denotes the interaction effect of two variables. Only inclusion of the interaction terms increased variance explained by the fixed effects, which was mainly due to the BDI*paradigm interaction. The variance explained by psychological factors in Models 3 and 5 was determined using Models 6 and 7, respectively (see Appendix A for a full breakdown of fixed effects variance). Note that the figure only illustrates the variance explained by the fixed effects and the residual inter-individual variance. The remaining variance is unexplained and may be due, e.g., to between-session differences.

**Table 1 brainsci-12-00623-t001:** Cross-sectional analysis (linear regression, Model 1, *n* = 126). Multiple R^2^ = 4.1%, *p* = 0.757. CS_Temp_ = conditioning stimulus temperature in °C. Sex and paradigms compared to a reference (male and 30 s heat/60 s cold, respectively). Paradigm 1 = heat 60 s/cold 90 s, Paradigm 2 = electrical/cold 120 s. CS_Temp_ = conditioning stimulus temperature in °C, BDI = Beck’s Depression Inventory score, STAI Trait = State-Trait Anxiety Inventory score (trait subscale), PCS = Pain Catastrophizing Scale score.

Predictor	Estimate	*p*-Value
CSTEMP	1.35	0.085
AGE	−0.07	0.840
SEX	1.35	0.817
BDI	−0.14	0.861
STAI TRAIT	−0.08	0.821
PCS	−0.30	0.289
PARADIGM 1	3.43	0.753
PARADIGM 2	5.05	0.442

**Table 2 brainsci-12-00623-t002:** Repeated measures analysis (mixed Models 2 and 3, 54 participants, 118 observations). Model 2: REML criterion at convergence = 1015.8. Model 3: REML criterion at convergence = 1014.7. *p*-values were obtained by Wald’s chi-square test on Models 2 and 3. Sex and paradigm compared to a reference (male and electrical/120 s cold, respectively). Significant effects are marked in bold. CS_Temp_ = conditioning stimulus temperature in °C, BDI = Beck’s Depression Inventory score, STAI Trait = State-Trait Anxiety Inventory score (trait subscale), PCS = Pain Catastrophizing Scale score.

Model	Predictor	Estimate	*p*-Value
**MODEL 2**	**CS_Temp_**	**1.54**	**0.001**
	Age	0.26	0.546
	Sex	−1.86	0.700
	Paradigm	−8.47	0.085
	Repeat	−3.07	0.190
**MODEL 3**	**CS_Temp_**	**1.55**	**0.002**
	Age	0.19	0.679
	Sex	−1.57	0.755
	BDI	−0.48	0.599
	STAI Trait	0.02	0.965
	PCS	0.00	0.999
	Paradigm	−7.60	0.197
	Repeat	−3.04	0.195

## Data Availability

Data is available upon reasonable request from the corresponding author.

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
