# Peer review of "The Contribution of Psychological Factors to Inter-Individual Variability in Conditioned Pain Modulation Is Limited in Young Healthy Subjects"

_brainsci, 2022, doi:10.3390/brainsci12050623_

Round 1

Reviewer 1 Report

This paper will be a good addition to the existing literature of CPM and inter-individual differences in CPM. The authors have done a good job exploring the topic of CPM further. There are certainly major strengths and limitations of the paper which the authors have correctly identified and discussed. The paper will be of general interest to readers exploring this field. 

The paper is a continuation of the authors/groups’ previous work on this topic published in the same journal, and there is no specific addition to the CPM methodology (except the inclusion of new stats to analyze the data). One suggestion to probably enhance the readability of the manuscript, would be for the authors to consider including a separate section called “Future work” that would guide the field/readers towards future investigation in this topic of research. Here the authors could expand/discuss, say for example, on the topic of potential influence of other psychological factors on CPM, such as attention, distraction, stress, nature and difficulty of CPM task, and the environmental demands (emotional arousal) implicated in moderating the interruptive function of pain. Another suggestion would be to categorize the strengths and limitations as a separate section

Author Response

Dear Reviewer,

We would like to thank you for your valuable feedback and comments on our manuscript. We believe that they have helped in improving the quality of our manuscript and made it more readable. Please find the following revisions addressing your comments in the attached manuscript:

Comment 1: One suggestion to probably enhance the readability of the manuscript, would be for the authors to consider including a separate section called “Future work” that would guide the field/readers towards future investigation in this topic of research. Here the authors could expand/discuss, say for example, on the topic of potential influence of other psychological factors on CPM, such as attention, distraction, stress, nature and difficulty of CPM task, and the environmental demands (emotional arousal) implicated in moderating the interruptive function of pain.

Response 1: We have added a sub-section entitled “Future direction” discussing possible avenues of future inquiry and suggesting further experiments (line 274 ff.). This sections includes discussion on other potential influences which read as follows (lines 275-285):

Several additional factors could contribute to inter-individual differences in CPM: First,  there may be other psychological factors, transient or not, that could influence the CPM effect. For example, active cognitive strategies have been shown to influence descending pain inhibition [32], [33], and intrinsic attention to pain, i.e. an individual’s innate tendency to attend to painful stimuli, is related to CPM [34]. Stressful tasks have also shown to inhibit the CPM effect [35]. Expectations towards the direction and magnitude of CPM may be individually different and can affect CPM magnitude [36]. Second, allelic differences in certain genes have an effect on both pain perception and CPM [37], [38]. Third, Ibancos-Losada et al. [39] suggested that individual differences in perceived unpleasantness of certain pain modalities over others may influence their CPM effect. Fourth, cardiovascular reactivity to pain may also affect CPM [40].

As well as lines 309-313:

Moreover, our present and previous [9], [17] data show that CPM paradigm has an effect on CPM magnitude and also on the relation between CPM and psychological factors. Future studies should take this into account and either use a single paradigm, or ideally compare multiple paradigms, including paradigms not examined in this study e.g. pressure pain.

Comment 2: Another suggestion would be to categorize the strengths and limitations as a separate section

Response 2: We have added a sub-section entitled “Strengths and limitations” in the discussion (Line: 404 ff.)

Reviewer 2 Report

The conclusion “The psychological factors depression, anxiety and pain catastrophizing did not make a significant contribution to explain inter-individual variance in the CPM effect of healthy young subjects” is not supported by the result as there is no group with depression/anxiety/pain catastrophizing has been included in the study. The authors have studied healthy subjects with no sign of psychological distress. The study design and material and methods are poorly described.

Author Response

Dear Reviewer,

Thank you for your comments on our manuscript, an updated version of which you will find attached. Please find the revisions therein, a summary of the changes responding to your comments are below:

Comment 1: The conclusion “The psychological factors depression, anxiety and pain catastrophizing did not make a significant contribution to explain inter-individual variance in the CPM effect of healthy young subjects” is not supported by the result as there is no group with depression/anxiety/pain catastrophizing has been included in the study. The authors have studied healthy subjects with no sign of psychological distress.

Response 1: Our aim was to investigate the influence of psychological factors on CPM within a healthy population. We have included clarification in the abstract and discussion that our findings pertain to healthy individuals only, which read:

Line 16: “In a previous analysis we demonstrated that in young, healthy participants inter-individual differences account for about one third of CPM variance, with age and sex together explaining only 1%”

Line 29: “These results show that the contribution of psychological factors to inter-individual CPM differences in healthy participants is limited”

Line 212: “This follow-up investigation shows that in young healthy subjects.”

We also added why an association between psychological factors and CPM in healthy people are to be considered a baseline (lines 68-70): “In addition, discerning the contribution of psychological factors to inter-individual CPM variance in a healthy population may establish a normative baseline to which their effect in chronic pain populations can be compared.”

 We discuss the need to investigate this association in chronic pain with and without elevated psychological scores, as well as pain-free individuals with elevated psychological scores(lines 297-304): “However, the situation is complex. Chronic pain patients often have psychological comorbidities, resulting in elevated scores for depression, anxiety and/or catastrophizing [3]–[5], and they also exhibit a relationship between these scores and  increased pain perception and/or a reduced CPM effect  [41]–[43]. Therefore, to dissect the relation between CPM, psychological factors and chronic pain, at least three different groups of patients will have to be compared: patients with chronic pain but without psychological comorbidity, patients with increased depression, anxiety and/or catastrophizing scores but without chronic pain, and patients with chronic pain and psychological comorbidities.”

Comment 2: The study design and material and methods are poorly described.

Response 2: We believe that have provided a sufficient description of the analysis methods employed in our study to comprehend our findings and referenced our previous publication, which provides a detailed description of the CPM procedure used to obtain the data and upon which this manuscripts build upon, at all relevant points.

Reviewer 3 Report

Overall comment

The work is very interesting and helpful in filling gaps in the literature. I think the introduction and discussion should be expanded to enhance this study.

Since CPM is interesting in relation to chronic pain, I would add something related to this condition even though the subjects recruited in the study are healthy. Since it is necessary to have normative data on healthy subjects and possible associations of CPM with (modifiable) psychological factors to assess whether CPM may be a biomarker for pain conditions, and possibly whether modifiable variables may have an effect on it. Authors should provide a more specific definition such as ‘’ Conditioned pain modulation (CPM) is a psychophysical experimental measure of the endogenous pain inhibitory pathway in humans: the “pain inhibits pain” phenomena’’- Conditioned pain modulation (CPM) is a quantitative sensory test commonly used to assess the functionality of endogenous pain inhibition in the central nervous system. During this paradigm, a nociceptive (test) stimulus is administered in the absence and after/during the application of a second painful (conditioning) stimulus, which is applied in a remote region of the body. The evaluation of pain provoked by the test stimulus can be done either at the same time (parallel paradigm) or after the conditioning stimulus (CS) has been withdrawn (sequential paradigm) (from 10.1097/j.pain.0000000000000689).

I would specify further that in healthy subjects and in subjects with chronic pain this neurophysiological mechanism works differently and possible explenations.

I would specify in the discussions that it may be that psychological traits were not significant findings precisely because they are healthy participants. In fact, it appears that CPM is a biomarker that helps discriminate between healthy subjects and those with chronic pain. 10.1097/j.pain.0000000000001664

Also, this work provides further evidence about the role of the modality of stimulation, the type of pain, and the stimulation site which appear to be critical variables that influenced the pattern of results.

Also, in the introduction and in the discussion it would be helpful to specify that this kind of evidence can be useful for providing normative data that would allow the use of the CPM response for diagnostic purposes and investigating whether CPM can be a prognostic biomarker of future pain (Yarnitsky D, Crispel Y, Eisenberg E, Granovsky Y, Ben-Nun A, Sprecher E, Best LA, Granot M. Prediction of chronic post-operative pain: pre-operative DNIC testing identifies patients at risk. PAIN 2008;138:22–8.)

Revision

line 50: which traits? Personality traits?

Line 51: how do they influence the perception of pain? let's specify the direction, ad we can say that they are factors associated with greater perceived pain intensity. Although there are some contradictory results doi: 10.3390/brainsci11010011 about the association of pain catastrophizing and pain intensity, and about the associations between depression and pain severity (https://doi.org/10.1186/1471-2296-10-54). Specifying the contradictory nature of the results and the lack of evidence on CPM might help highlight the gap in the literature that this article aims to fill. In addition, this work could also be useful in supporting the lack of association with catastrophizing.

I would also suggest including some evidence related to chronic pain conditions, where we detect associations between these factors and perceived pain intensity. (1) 10.3390/brainsci10100685  ; 2) Severeijns, Rudy M.Sc.*; Vlaeyen, Johan W.S. Ph.D.; van den Hout, Marcel A. Ph.D.; Weber, Wim E.J. Ph.D. Pain Catastrophizing Predicts Pain Intensity, Disability, and Psychological Distress Independent of the Level of Physical Impairment, The Clinical Journal of Pain ; 3) https://doi.org/10.1002/art.21865 ; 4) https://doi.org/10.1016/S0033-3182(10)70673-3)

line 208: I would add if there are any articles in which these variables were not found to be associated (pain intensity, pain perception, and psychological factors) to support your evidence

Author Response

Dear Reviewer,

We would like to thank you for your valuable feedback and comments on our manuscript. We believe that they have helped in improving the quality of our manuscript and made it more readable. Please find the following revisions addressing your comments in the attached manuscript:

Comment 1: Since CPM is interesting in relation to chronic pain, I would add something related to this condition even though the subjects recruited in the study are healthy. Since it is necessary to have normative data on healthy subjects and possible associations of CPM with (modifiable) psychological factors to assess whether CPM may be a biomarker for pain conditions, and possibly whether modifiable variables may have an effect on it. Authors should provide a more specific definition such as ‘’ Conditioned pain modulation (CPM) is a psychophysical experimental measure of the endogenous pain inhibitory pathway in humans: the “pain inhibits pain” phenomena’’- Conditioned pain modulation (CPM) is a quantitative sensory test commonly used to assess the functionality of endogenous pain inhibition in the central nervous system. During this paradigm, a nociceptive (test) stimulus is administered in the absence and after/during the application of a second painful (conditioning) stimulus, which is applied in a remote region of the body. The evaluation of pain provoked by the test stimulus can be done either at the same time (parallel paradigm) or after the conditioning stimulus (CS) has been withdrawn (sequential paradigm) (from 10.1097/j.pain.0000000000000689).

Response 1: We have expanded our explanation of CPM in the introduction (lines 35-41): “Conditioned pain modulation (CPM) describes a phenomenon of human endogenous pain inhibition thought to be the psychophysical equivalent to the “diffuse noxious inhibitory controls” (DNIC) described in animal experiments [1]. During CPM testing, a noxious test stimulus (TS) is presented in parallel with, or directly after, a heterotopic noxious conditioning stimulus (CS), with the underlying principle being summarized as “pain inhibits pain” [2]. When presented with the CS, the TS is perceived as less painful compared to presentation without CS [1].”

Comment 2: I would specify further that in healthy subjects and in subjects with chronic pain this neurophysiological mechanism works differently and possible explenations.

Response 2: We have specified in the discussion that CPM differs between healthy and chronic patients in lines 297-299:

“However, the situation is complex. Chronic pain patients often have psychological comorbidities, resulting in elevated scores for depression, anxiety and/or catastrophizing [3]–[5], and they also exhibit a relationship between these scores ad increased pain perception and/or a reduced CPM effect  [41]–[43].”

Comment 3: I would specify in the discussions that it may be that psychological traits were not significant findings precisely because they are healthy participants. In fact, it appears that CPM is a biomarker that helps discriminate between healthy subjects and those with chronic pain. 10.1097/j.pain.0000000000001664

Response 3: We have added this as possible explanation in lines 290-293:

“In addition, it is possible that we found little relation between psychological factors and CPM effect because our study population was healthy. CPM may not be affected by psychological factors if they are within a fairly narrow and low range, as they were within our study.”

Comment 4: Also, in the introduction and in the discussion it would be helpful to specify that this kind of evidence can be useful for providing normative data that would allow the use of the CPM response for diagnostic purposes and investigating whether CPM can be a prognostic biomarker of future pain (Yarnitsky D, Crispel Y, Eisenberg E, Granovsky Y, Ben-Nun A, Sprecher E, Best LA, Granot M. Prediction of chronic post-operative pain: pre-operative DNIC testing identifies patients at risk. PAIN 2008;138:22–8.)

Response 4: We have specified that our findings can be taken as a baseline useful for comparison in future studies in clinical populations in lines 68-70:

“In addition, discerning the contribution of psychological factors to inter-individual CPM variance in a healthy population may establish a normative baseline to which their effect in chronic pain populations can be compared.” And in lines 293-295:

“However, this is a necessary first step in investigating the relation between psychological factors and CPM, also to establish a normative baseline for a healthy population”

We have also expanded in our discussion the need to investigate healthy and chronic pain subjects, as well as patients without chronic pain but with elevated psychological scores separately in order to further classify the influence on CPM in lines 300-307:

“Therefore, to dissect the relation between CPM, psychological factors and chronic pain, at least three different groups of patients will have to be compared: patients with chronic pain but without psychological comorbidity, patients with increased depression, anxiety and/or catastrophizing scores but without chronic pain, and patients with chronic pain and psychological comorbidities. Also, chronic pain populations can be very heterogeneous regarding type and cause of chronic pain. Limiting investigation to one of the major types of chronic pain known to be associated with reduced CPM effect might be a good starting point.”

Comment 5: line 50: which traits? Personality traits?

Response 5: We have omitted the word “traits” and clarified that we mean the psychological factors investigated in the manuscript in line 56-58:

“Psychological factors might explain part of these residual inter-individual differences. Pain perception has been shown to increase with higher scores of anxiety, depression and pain catastrophizing [10]–[12].”

Comment 6: Line 51: how do they influence the perception of pain? let's specify the direction, ad we can say that they are factors associated with greater perceived pain intensity. Although there are some contradictory results doi: 10.3390/brainsci11010011 about the association of pain catastrophizing and pain intensity, and about the associations between depression and pain severity (https://doi.org/10.1186/1471-2296-10-54). Specifying the contradictory nature of the results and the lack of evidence on CPM might help highlight the gap in the literature that this article aims to fill. In addition, this work could also be useful in supporting the lack of association with catastrophizing.

Response 6: We have included references that reflect the contradictory results on CPM and psychological factors in lines 58-59 “However, discrepancies in the literature exist, with other studies finding no such association [13]–[15].”

and lines 227-229: “However, it must be mentioned that other studies find no relationship between pain perception and anxiety [14], catastrophizing [15] and depression [13] in healthy individuals.”

Comment 7: I would also suggest including some evidence related to chronic pain conditions, where we detect associations between these factors and perceived pain intensity. (1) 10.3390/brainsci10100685  ; 2) Severeijns, Rudy M.Sc.*; Vlaeyen, Johan W.S. Ph.D.; van den Hout, Marcel A. Ph.D.; Weber, Wim E.J. Ph.D. Pain Catastrophizing Predicts Pain Intensity, Disability, and Psychological Distress Independent of the Level of Physical Impairment, The Clinical Journal of Pain ; 3) https://doi.org/10.1002/art.21865 ; 4) https://doi.org/10.1016/S0033-3182(10)70673-3)

Response 7: We have added the section “Future directions” in which we expand on possible next steps in helping to elucidate the relationship between psychological factors and CPM in chronic pain conditions in lines 297-300:

“Chronic pain patients often have psychological comorbidities, resulting in elevated scores for depression, anxiety and/or catastrophizing [3]–[5], and they also exhibit increased pain perception and a reduced CPM effect [16], [41]–[43].”

Comment 8: line 208: I would add if there are any articles in which these variables were not found to be associated (pain intensity, pain perception, and psychological factors) to support your evidence

Response 8: See our answer to comment 6 above

Round 2

Reviewer 2 Report

Thank you for revised manuscript. It has been improved now.